# Determinants of depression among ever-married adolescent girls in Bangladesh: Evidence from the Bangladesh Adolescent Health and Wellbeing Survey 2019–2020

**Mehedi Hasan**[1]*, **Md Al Amin**[2]

**1** Research Monitoring and Evaluation Division, Room to Read, Dhaka, Bangladesh, **2** Health Systems and Population Studies Division, International Centre for Diarrheal Disease Research, Bangladesh (icddr,b), Dhaka, Bangladesh

\* mehdi631@yahoo.com

**Data Availability Statement:** All data are in the manuscript and/or Supporting information files.

**Funding:** The author(s) received no specific funding for this work.

## Abstract

### Background

Common mental health disorders in early life represent a major challenge and these conditions become more complicated and substantial during the development period of adolescence. Despite the global concern over the increasing prevalence of mental health issues among adolescents, it remains largely a neglected area of research and health policy in Bangladesh, where the burden of child marriage is significantly higher. This study aimed to investigate the prevalence and factors associated with depression among married adolescent girls in Bangladesh.

### Methods

The study utilized the data from first-ever Bangladesh Adolescent Health and Wellbeing Survey (BAHWS), conducted in 2019–2020. A total of 1,693 married adolescent girls were included in the final analysis. Depressive symptoms were measured using the standardized Patient Health Questionnaire-9 (PHQ-9) tool. Descriptive statistics were applied to assess the prevalence of depression, while bivariate analysis was done to measure the significance of the variables. Furthermore, logistic regression was used to examine the association between any form of depression and selected covariates.

### Results

The overall prevalence of mild to severe levels of depression among the participants was 53.1% (Mild: 40.3%; Moderate: 9.9%; Moderately severe: 2.3%; Severe: 0.6%). In the multivariable logistic regression model, it was found that adolescents from richest wealth quintile were 35% less likely to suffer from depression (AOR = 0.65; 95% CI = 0.45–0.92; P value = 0.02). Additionally, physical violence (AOR = 1.55; 95% CI = 1.14–2.09; P value = 0.004), sexual harassment (AOR = 1.50; 95% CI = 1.11–2.01; P value = 0.007), social bullying

**Competing interests:** The authors have declared that no competing interests exist.

(AOR = 2.25; 95% CI = 1.73–2.91; P value < .001), and cyberbullying (AOR = 1.75; 95% CI = 1.10–2.77; P value = 0.01) were associated with depression.

## Conclusions

This study demonstrated that more than half of the married adolescents suffer from mild to severe levels of depression, while any form of violence and harassment significantly increases their risk of depression. Therefore, a more inclusive policy is needed, engaging with communities and local stakeholders, to lay out key actions and intervention pathways to address the issue of violence against married adolescent girls as the extent and spectrum of violence continue to evolve.

## Introduction

Adolescence is considered as a transitional period when a human being transforms from childhood to adulthood through physical, intellectual, psychological, and social challenges. The World Health Organization (WHO) defines the adolescent period as ranging from 10–19 years of age [1]. During adolescence, individuals undergo multifaceted changes that occur rapidly and concurrently. Since these changes occur abruptly and in tandem, therefore the adolescent period is labeled as a period of vulnerability and susceptibility to any form of physical and mental health disorder including depression [1]. According to WHO, depression is a mental health condition characterized by persistent loss of interest in activities and a negative impact on all aspects of life, including relationships with family, friends, and the community. It is a prevalent mental health disorder, affecting approximately 264 million people worldwide, and leading contributor to the global burden of disease [2].

Globally, the prevalence of mental health disorders among adolescents is significantly increasing. A systematic review and meta-analysis showed the pooled prevalence of mental health disorders to be 13.4% among adolescents and children [3]. Notably, depression is a significant risk factor for poor health outcomes, including increased risk of suicide, unhealthy lifestyle behaviors, and diminished social well-being among adolescents [4, 5]. In addition, early-life depression is also associated with poor educational attainment [6].

In Bangladesh, several studies have highlighted the higher prevalence of depression among adolescents. For example, one study found that 36.6% of adolescents in urban and semi-urban areas suffer from depressive symptoms, with a higher prevalence among girls compared to boys [6]. Another study reported that 19% of male and 30% of female school-going adolescents in Dhaka city have depressive symptoms [7, 8]. A cross-sectional study reported that the prevalence of depressive symptoms is 27% among adolescent girls and 22% among boys [9]. However, studies conducted in Bangladesh were mostly on a small scale and mainly focused on unmarried adolescent boys and girls, reflecting a huge knowledge gap regarding the depression status and potential risk factors of married adolescents.

Married adolescent women are more likely to experience social isolation, limited autonomy, limited access to mobility, and exposure to mass media, all of which increase their vulnerability to poor mental health [10]. Due to culturally grounded gender roles in Bangladesh, married adolescent women are found to have little or no decision-making autonomy and they are often hesitant to communicate with their husbands and family members regarding their health conditions and utilization of reproductive health services [11–13]. A systematic review and meta-analysis identified social isolation as a potential risk factor for depression and other mental health conditions among Bangladeshi married adolescents [14]. Similarly, the

adolescents' lack of autonomy in health-seeking behavior, purchasing ability, and food consumption also increase the risk of depression [15]. Early marriage is a prevalent issue in Bangladesh, with 51% of girls married off before the age of 18, representing the highest prevalence of child marriage in the Southeast Asian region [16]. Evidence indicates that early marriage increases the risk of Intimate Partner Violence (IPV) [17], which in turn a potential risk factor for poor mental health outcomes among the married women [18, 19]. A systematic review investigated a wide range of health consequences associated with early marriage, including mental health disorders, poor maternal health status, risk of IPV and poor health seeking behavior [20]. Another study reported that, 61% of married adolescent in rural Bangladesh experienced depressive symptoms during the Covid-19 pandemic [21]. Suicidal behavior is common among married adolescent women. For instance, a study reported 22-fold greater risk of suicidal behavior among married adolescents compared to never married adolescent in Bangladesh [22]. Moreover, the psychiatric factors like depression also elevated the risk of suicidal ideation [23]. Despite these consequences, the mental health of married adolescent girls in Bangladesh is largely ignored and no study systematically analyzed the socio-demographic determinants of depression to help in designing the evidence-based interventions to reduce the burden of this condition. However, the National Strategy for Adolescent Health 2017–2030 emphasizes the adolescents' mental health status by documenting the available epidemiological evidence in order to improve the mental well-being of Bangladeshi adolescents and to achieve the Sustainable Development Goals (SDGs) [24].

Given the substantial knowledge gap about ever-married adolescent girls' mental health status and its determinants, this study seeks to address this gap and assist policymakers, government and relevant stakeholders in making evidence-based informed decisions in order to improve mental health status of married adolescents girls. The specific objective of this study is to investigate the prevalence and potential socio-demographic determinants of depression among married adolescent women in Bangladesh, utilizing secondary data.

## Materials and methods

### Data sources

The Bangladesh Adolescent Health and Well-being Survey (BAHWS) 2019–20 is the first-ever nationally representative survey that comprehensively collected data on the health and well-being status of adolescents (15–19 years). This gender integrated survey includes ever-married female, never-married female and never-married male. In this study, we leveraged this survey data to quantify the depression status and associated factors among married adolescent girls [25]. Between July 2019 and January 2020, the National Institute of Population Research and Training (NIPORT) and the Health Education and Family Welfare Division of the Ministry of Health and Family Welfare (MOHFW) conducted the survey with technical assistance from the United States Agency for International Development (USAID). The primary objective of this survey was to determine the extent of the health and well-being status of male and female adolescents ages 15–19 years, to address the knowledge gap and inform policy decisions.

### Survey design

The Adolescent Health and Well-being Survey (BAHWS) 2019–20 utilized a two-stage stratified sampling technique. In the 1st stage, a list of Primary Sampling Units (PSUs) was prepared using the 2011 Population and Housing Census. Subsequently, a predetermined number of PSUs was selected from each stratum by applying the probability proportional to size (PPS) sampling method. In the 2nd stage, within the selected PSU, a comprehensive household listing was performed to provide a sample frame for the selection of households. The PSU of the

survey is an Enumerator Area (EA) that covers on average 100 households, consisting of both rural and urban settings. A total of 728 EAs were randomly selected from the sample frame. Thereafter, from each selected EAs, 100 households were selected for interviews, resulting in a total of 72,800 (Rural: 51,400, Urban: 21,400) households. Of them, interviews were conducted in 67,093 households with a response rate of 97.7%. Within those households, the survey interviewed 18,249 adolescents. Among them, 4,926 were ever-married female, 7,800 were unmarried female, and 5,523 were unmarried male adolescents.

## Study population

In this study, we considered only ever-married women to assess the prevalence and risk factors associated with depression. A total of 4,926 ever-married women aged 15–19 years were interviewed in the survey, achieving a success rate of 98%. However, depression status was assessed in 2,487 women out of 4,926 respondents. Of these, we purposively excluded the depression status of 383 pregnant women from the analysis as depressive symptoms are associated with pregnancy and early childbearing period, which may somewhat mislead the true depression status and associated risk factors [26]. In addition, 394 observations of depression status were further excluded due to the missing information on harassment and violence indicators, yielding a sample size of 1,693 for the final analysis. Since the missing data for these indicators are random and unrelated to the outcome of depression, a listwise deletion process was applied to systematically exclude 394 observations related to depression status. The details of the inclusion and exclusion criteria for sampling are presented in Fig 1.

## Outcome measures

The nine-item Bangla version Patient Health Questionnaire– 9 (PHQ-9) was used to assess depressive symptoms among the respondents [27]. The PHQ-9 is a simple, effective, and reliable tool for screening and evaluation of depression status at the population level (Cronbach's $\alpha = 0.77$) [28]. The PHQ-9 tool has a sensitivity of 89.5% and a specificity of 78.8% for detecting and investigating depressive symptoms among adolescents [29]. Moreover, PHQ-9 was found to have similar validity and reliability (Cronbach's $\alpha = 0.83$ and mean inter-item correlation = 0.34) when used to assess the depression status of Bangladeshi adolescents [6]. Using

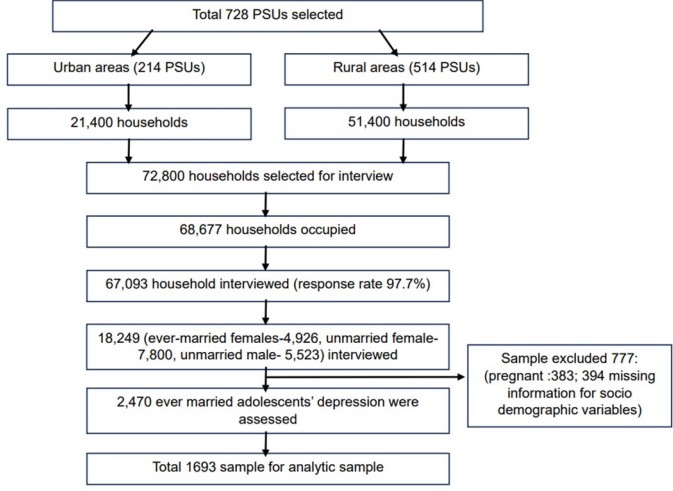

**Fig 1. Sampling process.**

the PHQ-9, the study participants were asked: "how often they experienced nine specific symptoms related to depression in the last 2 weeks preceding the survey" to assess the prevalence of depressive symptoms. Each symptom consisted of a four-point Likert scale: not at all (score = 0), several days (score = 1), more than half of the days (score = 2), and nearly every day (score = 3) [30]. The scores of the individual participants for all nine symptoms are combined and categorized into five levels of depression: none or minimal depression (score 0–4), mild depression (score 5–9), moderate depression (score10–14), moderately severe depression (score 15–19) and severe depression (score 20–27). Moreover, a PHQ-9 score between 5–27 was categorized as having 'any depression', while between 0–4 was considered as 'no or nominal depression'.

## Covariates measure

The covariates included in this study were selected based on previous literature reporting the determinants of depression in Low- and middle-income countries (LMICs), including Bangladesh [4, 9, 21]. In addition, the study was guided by a conceptual framework to select the covariates and other risk factors. The framework considered socio-demographic factors and exposure to violence indicators for predicting depressive symptoms among married adolescents. The socio-demographic factors included: administrative divisions (Barishal, Chattogram, Dhaka, Khulna, Rajshahi, Rangpur, Sylhet, and Mymensingh); place of residence (urban, rural); wealth status (poorest, poorer, middle, richer, richest) based on pre-set cutoffs; age of the participants (15–17, 18–19); education level (primary incomplete, primary complete, secondary incomplete, secondary complete); mass media exposure (yes, no); and social media exposure (yes, no). On the other hand, violence-related factors comprised physical violence and sexual harassment, while social bullying and cyberbullying were considered as harassment. The BAHWS 2019–20 survey used a set of items for each domain to measure the violence and harassment experienced by the respondents in the 12 months preceding the survey [25]. Physical violence was assessed with five items (slapped, punched, kicked, choked threatened, or attacked with a knife). Sexual harassment was assessed based on whether the respondent was treated in a vulgar way; encountered a sly whistle; got touched; forced to watch obscene photos or videos and other experiences related to violence. Social bullying was assessed based on items: cursing or passing mean comments; being blamed as a liar; receiving verbal or written threats and being socially excluded, whereas cyberbullying was assessed based on whether the respondent was harassed or bothered through a mobile phone or internet. The response categories for each item were categorized as 'yes' if the respondent experienced the phenomenon and 'no' if she didn't.

## Statistical analyses

Data were analyzed using STATA version 13. The survey data, initially in Microsoft Excel format, were then imported into STATA for cleaning and analysis. Descriptive statistics (e.g., frequencies, percentages) and bivariate analyses (i.e., chi-square tests) were conducted to assess the prevalence of depression and its level of significance. Logistic regression (both unadjusted and adjusted models) was performed, yielding odds ratios and their 95% confidence intervals, to examine associations between depression and selected independent variables. Analyses were univariable that yielding crude odds ratios (COR), followed by multivariable analyses, where only significant factors from the univariable analyses were included together, yielding Adjusted Odds Ratios(AOR) for depression, independent of other covariates. Sampling weights were applied in regression analyses to ensure the representativeness and robustness of

the findings. Variables were considered statistically significant if the two-sided p-value was less than 0.05.

## Results

### Background characteristics of the respondents

A total of 1,693 respondents were included in the final analysis, with a mean age of 17.6 years (SD = 1.24), ranging from 15 to 19 years. Descriptive statistics for all variables are presented in Table 1. The majority of the respondents are from rural areas (77.2%); are currently not attending school (83.6%) and did not complete secondary education (54%). The wealth quintile was evenly distributed among the respondents. For instance, 20% were from the poorest quintile, 21% from the poorer quintile, 22.6% from middle quintile, 21% from richer quintile, and 15% from the richest quintile. Regarding media exposure, 66% had exposure to mass media and 29.8% to social media exposure. In terms of violence and harassment, 19.6% reported physical violence, 16.7% reported sexual harassment, 30% reported social bullying, and 7% experienced cyberbullying in the 12 months preceding the survey (Table 1).

### Prevalence of depression

The prevalence of depression among ever-married female adolescents was 53.1%. The distribution of depression levels according to the PHQ-9 scale is as follows: 46.9% of the respondents were diagnosed with no or minimal depression, 40.3% with mild depression, 9.9% with moderate depression, 2.3% with moderately severe depression, and 0.6% with severe depression (see Fig 2).

### Prevalence of any depression status by selected covariates

Table 2 presents the bivariate distribution of any depression with selected independent variables. Six variables were found to be significant with a p-value <0.05: wealth quintile, current school attendance, physical violence, sexual harassment, social bullying, and cyberbullying. The proportion of depressive symptoms was highest among women from the poorest wealth quintile (59%) and lowest among those from the richest wealth quintile (46%). Additionally, depressive symptoms were lower among those currently attending school (48%), compared to those not participating in school (54%). Depressive symptoms were more prevalent among the girls who experienced violence and harassment in the last 12 months preceding the survey. For instance, married adolescents who experienced physical violence and sexual harassment were more likely to be depressed compared to those who did not experience any violence. Likewise, the proportion of depression was 72% among girls who experienced physical violence, compared to 48% among those who did not. Similarly, 68% of the respondents who experienced sexual harassment exhibited depressive symptoms, compared to 50% of girls who had no experience of this kind of harassment. On the other hand, the proportion of depression was higher among girls who experienced social bullying (72%) and cyberbullying (75%), compared with those who didn't face any bullying (Table 2).

### Factors associated with depressive symptoms

The results of univariable and multivariable logistic regression analyses are presented in Table 3. Depression was significantly associated with educational status, wealth quintile, experience of physical violence, sexual harassment, and social and cyberbullying in the univariable analysis.

**Table 1. Descriptive statistics of the study population.**

| Variables | Total N = 1,693 | |
|---|---|---|
| | **n** | **%** |
| **Sociodemographic variables** | | |
| **Age** | | |
| 15–17 | 711 | 42.0 |
| 18–19 | 982 | 58.0 |
| **Place of residence** | | |
| Rural | 1,307 | 77.2 |
| Urban | 386 | 22.8 |
| **Division** | | |
| Barishal | 105 | 6.2 |
| Chattogram | 334 | 19.7 |
| Dhaka | 359 | 21.2 |
| Khulna | 204 | 12.1 |
| Rajshahi | 272 | 16.1 |
| Rangpur | 203 | 12.0 |
| Sylhet | 67 | 4.0 |
| Mymensingh | 149 | 8.8 |
| **Wealth quintile** | | |
| Poorest | 343 | 20.3 |
| Poorer | 361 | 21.3 |
| Middle | 382 | 22.6 |
| Richer | 352 | 20.8 |
| Richest | 255 | 15.1 |
| **Educational status** | | |
| Primary incomplete | 204 | 12.1 |
| Primary complete | 162 | 9.6 |
| Secondary incomplete | 916 | 54.1 |
| Secondary complete | 411 | 24.3 |
| **Currently attending school** | | |
| Yes | 277 | 16.4 |
| No | 1,416 | 83.6 |
| **Exposure to mass media** | | |
| Yes | 1,118 | 66.0 |
| No | 575 | 34.0 |
| **Exposure to social media** | | |
| Yes | 504 | 29.8 |
| No | 1,189 | 70.2 |
| **Violence and harassment-related variables** | | |
| **Physical violence** | | |
| Yes | 332 | 19.6 |
| No | 1,361 | 80.4 |
| **Social bullying** | | |
| Yes | 508 | 30.0 |
| No | 1,185 | 70.0 |
| **Cyberbullying** | | |
| Yes | 118 | 7.0 |
| No | 1,575 | 93.0 |

(*Continued*)

**Table 1.** (Continued)

| Variables | Total N = 1,693 | |
|---|---|---|
| | **n** | **%** |
| **Sexual harassment** | | |
| Yes | 282 | 16.7 |
| No | 1,411 | 83.3 |

In multivariable analyses, where all significant factors from univariable analyses were entered together in the model, most associated factors retained significance but the effects were somewhat attenuated. Respondents from the richest wealth quintile were 35% less likely to be depressed compared to those from the poorest wealth quintile (AOR = 0.65; 95% CI = 0.45–0.92; P value = 0.02). Likewise, the odds of having depression were 55% higher among those who had experienced physical violence (AOR = 1.55; 95% CI = 1.14–2.09; P value = 0.004), 50% higher for sexual harassment (AOR = 1.50; 95% CI = 1.11–2.01; P value = 0.007), 2.25 times higher for those who experienced social bullying (AOR = 2.25; 95% CI = 1.73–2.91; P value <0.001), and 75% higher for the respondents who experienced cyber-bullying (AOR = 1.75; 95% CI = 1.10–2.77; P value = 0.01) (Table 3). However, although educational status was found to be significant in the univariable analysis, its effect was attenuated in the multivariable analysis and became insignificant (P value = >0.05).

## Discussion

To the best of our knowledge, this study is the first in Bangladesh to report the prevalence of and factors associated with depression using the first-ever nationally representative sample of married adolescent girls. Our findings found a high prevalence of depression among respondents across various socio-demographic and violence indicators. Depression was associated with wealth quintile, physical violence, sexual harassment, social bullying, and cyberbullying even after controlling for potential confounders.

Our study revealed that 53.1% of the respondents exhibited mild to severe depressive symptoms. This prevalence is lower compared to a previous study, which reported a depression rate of 61% [21]. The variation in prevalence rates can be attributed to differences in the nature

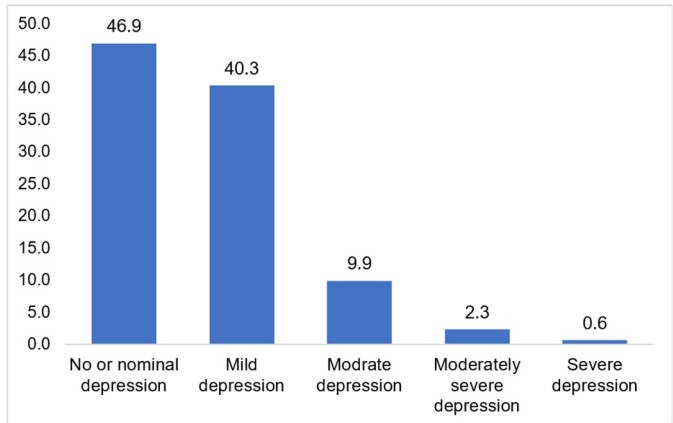

**Fig 2. Prevalence of levels of depression among ever-married adolescent girls.**

**Table 2. Association between any depression and selected independent variables: A bivariate analysis.**

| Variables | Any depression (%) | | P value |
|---|---|---|---|
| | **No** | **Yes** | |
| **Age** | | | |
| 15–17 | 45.9 | 54.2 | 0.462 |
| 18–19 | 47.7 | 52.3 | |
| **Place of residence** | | | |
| Rural | 45.9 | 54.2 | 0.640 |
| Urban | 47.2 | 52.8 | |
| **Division** | | | |
| Barishal | 41.0 | 59.1 | 0.240 |
| Chattogram | 48.5 | 51.5 | |
| Dhaka | 51.3 | 48.8 | |
| Khulna | 44.6 | 55.4 | |
| Rajshahi | 48.9 | 51.1 | |
| Rangpur | 44.3 | 55.7 | |
| Sylhet | 47.8 | 52.2 | |
| Mymensingh | 39.6 | 60.4 | |
| **Educational status** | | | |
| Primary incomplete | 42.7 | 57.4 | 0.273 |
| Primary complete | 42.0 | 58.0 | |
| Secondary incomplete | 48.1 | 51.9 | |
| Secondary complete | 48.2 | 51.8 | |
| **Currently attending school** | | | |
| Yes | 51.6 | 48.4 | 0.008 |
| No | 46.0 | 54.0 | |
| **Wealth quintile** | | | |
| Poorest | 40.8 | 59.2 | 0.008 |
| Poorer | 43.5 | 56.5 | |
| Middle | 50.8 | 49.2 | |
| Richer | 47.2 | 52.8 | |
| Richest | 53.7 | 46.3 | |
| **Exposure to mass media** | | | |
| Yes | 47.1 | 52.9 | 0.784 |
| No | 46.4 | 53.6 | |
| **Exposure to social media** | | | |
| Yes | 46.0 | 54.0 | 0.642 |
| No | 47.3 | 52.7 | |
| **Physical violence** | | | |
| Yes | 28.0 | 72.0 | <0.001 |
| No | 51.5 | 48.5 | |
| **Sexual harassment** | | | |
| Yes | 31.9 | 68.1 | <0.001 |
| No | 49.9 | 50.1 | |
| **Social bullying** | | | |
| Yes | 28.2 | 71.9 | <0.001 |
| No | 54.9 | 45.1 | |
| **Cyberbullying** | | | |

(*Continued*)

**Table 2.** (Continued)

| Variables | Any depression (%) | | P value |
|---|---|---|---|
| | No | Yes | |
| Yes | 24.6 | 75.4 | <0.001 |
| No | 48.6 | 51.4 | |

The p-value was significant at <0.05.

The cut-off of PHQ-9 score ≥5 was categorized as any depression for this analysis.

and context of the studies and the measurement scales used. For instance, the reference study mainly focused on rural areas and employed Depression, Anxiety, and Stress Scale (DASS-21), which contains 7 items for assessing depression, whereas our study used the PHQ-9 [21, 31]. Furthermore, the study was conducted during the COVID-19 pandemic period, and it is well established that the COVID-19 pandemic imposes a 2-3-fold increased risk of mental health disorders among adolescents in Bangladesh [32]. Consistent with our findings, a study reported a prevalence of 43% among unmarried adolescent girls using the PHQ-9 tool [9]. The high prevalence of depression in our study can be explained by different cultural norms and practices in Bangladesh. Cultural factors such as early marriage, IPV, inequality, poor subjective happiness, and limited mobility contributed to poor mental health outcomes among married women in Bangladesh [12]. For instance, early marriage limits educational attainment and financial independence, which exacerbates the feeling of hopelessness and depression among married adolescents in Bangladesh [33]. In addition, mental health issues are recognized as stigma, which prevents adolescents from seeking treatment and support [34].

As revealed in our study, the significant factors associated with depression were the lowest socio-economic status, experiences of physical violence, sexual harassment, social bullying, and cyberbullying. The association between low wealth quintile and depression is well documented. Socio-economic status is considered a protective factor and important social determents of health. A study found that individuals from low-income families are 49% more likely to experience depressive symptoms than those with high family income [35]. Moreover, financial strain was associated with an increased burden of depressive symptoms [36]. In contrast, higher socio-economic status facilitates quality health care and increased health knowledge resulting in a larger set of social and psychological resources that shape good health and well-being [37, 38]. These findings are consistent with our study, which shows that adolescents from the lowest wealth quintile were more likely to experience depressive symptoms than their richest counterparts. In LMICs, limited or no access to financial and household decision-making autonomy triggered poor mental health outcomes [39, 40].

The association between physical violence and depression is not uncommon. Previous studies have confirmed that any form of violence towards women leads to mental health problems [41, 42]. Violence often induces a loss of self-confidence among women, making them often more vulnerable to poor mental health conditions [19]. In the context of Bangladesh, adolescent wives often face limited mobility and are subject to restrictive social norms, early childbearing, and short birth intervals and these factors lead to increased risk of IPV [43]. On the other hand, IPV is considered one of the most common forms of physical and sexual violence that significantly increases the risk of mental disorders [44]. Our findings revealed that adolescents who experienced physical violence were more likely to develop depression, which is consistent with other studies conducted in Bangladesh [45]. This finding also supports a large-scale study conducted in India that identified an association between physical violence and

**Table 3. Logistic regression model for factors associated with any depression.**

| Variables | Any Depression | | | | | |
|---|---|---|---|---|---|---|
| | COR | (95% CI) | P value | AOR | (95% CI) | P value |
| **Socio-economic variables** | | | | | | |
| **Age** | | | | | | |
| 15–17 (Reference) | | | | | | |
| 18–19 | 1.08 | (0.88–1.30) | 0.46 | NA | NA | NA |
| **Place of residence** | | | | | | |
| Rural (Reference) | | | | | | |
| Urban | 1.06 | (0.84–1.32) | 0.64 | NA | NA | NA |
| **Division** | | | | | | |
| Barishal (Reference) | | | | | | |
| Chattogram | 0.74 | (0.47–1.14) | 0.18 | NA | NA | NA |
| Dhaka | 0.66 | (0.42–1.02) | 0.06 | NA | NA | NA |
| Khulna | 0.86 | (0.53–1.38) | 0.54 | NA | NA | NA |
| Rajshahi | 0.72 | (0.45–1.14) | 0.17 | NA | NA | NA |
| Rangpur | 0.87 | (0.54–1.40) | 0.57 | NA | NA | NA |
| Sylhet | 0.76 | (0.40–1.40) | 0.38 | NA | NA | NA |
| Mymensingh | 1.06 | (0.63–1.76) | 0.83 | NA | NA | NA |
| **Educational status** | | | | | | |
| Primary incomplete (Reference) | | | | | | |
| Primary complete | 0.91 | (0.58–1.41) | 0.68 | 0.90 | (0.56–1.42) | 0.65 |
| Secondary incomplete | 0.70 | (0.50–0.98) | 0.04 | 0.72 | (0.50–1.02) | 0.07 |
| Secondary complete | 0.71 | (0.49–1.02) | 0.07 | 0.83 | (0.55–1.23) | 0.35 |
| **Currently studying** | | | | | | |
| Yes (Reference) | | | | | | |
| No | 1.27 | (0.97–1.64) | 0.07 | NA | NA | NA |
| **Wealth quantile** | | | | | | |
| Richest | 0.59 | (0.38–0.87) | 0.00 | 0.65 | (0.45–0.92) | 0.02 |
| Richer | 0.67 | (0.53–1.10) | 0.01 | 0.84 | (0.61–1.15) | 0.29 |
| Middle | 0.77 | (0.50–0.98) | 0.09 | 0.74 | (0.54–1.01) | 0.06 |
| Poorer | 0.90 | (0.66–1.26) | 0.47 | 0.95 | (0.69–1.29) | 0.69 |
| Poorest (Reference) | | | | | | |
| **Mass media exposure** | | | | | | |
| Yes (Reference) | | | | | | |
| No | 1.03 | (0.84–1.25) | 0.78 | NA | NA | NA |
| **Social Media exposure** | | | | | | |
| Yes (Reference) | | | | | | |
| No | 0.95 | (0.77–1.17) | 0.64 | NA | NA | NA |
| **Violence and Harassment related variables** | | | | | | |
| **Physical violence** | | | | | | |
| Yes | 2.73 | (2.10–3.54) | <0.001 | 1.55 | (1.14–2.09) | 0.004 |
| No (Reference) | | | | | | |
| **Sexual harassment** | | | | | | |
| Yes | 2.12 | (1.61–2.78) | <0.001 | 1.50 | (1.11–2.01) | 0.007 |
| No (Reference) | | | | | | |
| **Social bullying** | | | | | | |
| Yes | 3.11 | (2.48–3.89) | <0.001 | 2.25 | (1.73–2.91) | <0.001 |
| No (Reference) | | | | | | |

*(Continued)*

**Table 3.** (Continued)

| Variables | Any Depression | | | | | |
|---|---|---|---|---|---|---|
| | COR | (95% CI) | P value | AOR | (95% CI) | P value |
| **Cyberbullying** | | | | | | |
| Yes | 2.90 | (1.88–4.45) | <0.001 | 1.75 | (1.10–2.77) | 0.01 |
| No (Reference) | | | | | | |

Note: COR: Crude Odds Ratio; CI: Confidence Interval; AOR: Adjusted Odds Ratio; NA: Not Applicable

The p value was significant at <0.05.

The cut-off of PHQ-9 score ≥5 was categorized as any depression for this analysis.

depression [46]. However, the potential pathways through which physical violence leads to depression and other mental health illnesses have not been well studied. However, one study reported physical violence directly triggered problematic behaviors in adolescents through expressive suppression and anger, which subsequently led to depressive symptoms [47]. Another study documented emotional distress, suicidal ideations, and suicidal attempts as potential pathways to develop depressive symptoms [48]. Similarly to physical violence, sexual harassment increased the likelihood of depression found in our study and these findings are aligned with the recently published studies conducted in LMICs [49, 50]. Notably sexual harassment is a significant stressor for victims, leading to social isolation and increasing the risk of any mental health condition [51]. In the context of Bangladesh, married adolescents are more likely to experience sexual violence by their husbands due to persistent cultural norms and behaviors [52]. Lack of autonomy in sexual behavior significantly mediates the association between sexual violence and depression among the adolescent [53]. Additionally, women who experienced sexual violence often suffer from Post-Traumatic Stress Disorder (PTSD), which, in turn, increases the risk of depression [51].

Another important finding in our study revealed that adolescent girls who experience social bullying were more likely to suffer from depressive symptoms. Social bullying is a common form of violence affecting school-going girls and adolescents in Bangladesh [54]. Evidence shows that social bullying exacerbates the risk of depressive symptoms and other mental health conditions among adolescent girls, which supports our findings [55]. Furthermore, victims of bullying had a threefold increased risk of tobacco use, self-harm, and lower job satisfaction in later life [56]. In addition, this form of harassment often leads to social isolation, traumatization, and identity crisis and these types of behaviors are subsequently posed mental health illness [57].

With technological advancement, cyberbullying has been evolving for the time being, and women are frequently harassed and victimized by this form of bullying. Our study unveiled that, married adolescent girls who experience internet-based harassment are more likely to suffer from depressive symptoms. Studies have explicitly revealed that a substantial number of adolescent girls experience cyberbullying or internet-based harassment [58, 59]. Furthermore, several studies have confirmed that mental health problems are associated with cyberbullying and internet-based violence among adolescent girls [60, 61] The widespread use of social networking platforms like Facebook, WhatsApp, and Viber has led to internet-based harassment among adolescent girls [62, 63]. Evidence shows that cyberbullying may be a contributing factor to substance use, which may further increase adolescents' susceptibility to mental health problems [64]. The higher rate of cyberbullying and internet-based violence among married adolescents could be attributed to their mental health status, through decreasing self-esteem

and problematic behavior, which is often associated with a higher rate of depressive symptoms [65, 66].

A notable association was observed between depression and educational attainment in the crude analysis, but the association no longer persisted in the multivariable analysis. Evidence suggests that years of schooling are associated with personal skills and employment, which further facilitates in reducing the onset of depressive symptoms [67]. Education facilitates access to employment, improves the quality of life, and enhances health knowledge, all of these lead to good health and well-being [37]. However, a systematic review and meta-analysis found no concrete evidence supporting the importance of education in lowering the onset of depression among adolescents, which is consistent with our findings [68].

## Limitations

We believe that our study is among the first of its kind that assess the association between socio-demographic status and depressive symptoms among married adolescent girls in Bangladesh, utilizing the country's first-ever nationally representative data on adolescent health and well-being. Despite this strength, our study has a few limitations. First, the depressive symptoms were self-reported rather than clinically diagnosed which may lead to a somewhat underestimation or overestimation of the true prevalence of depressive symptoms. Furthermore, there is a high probability of under-reporting the prevalence of violence and harassment due to recall bias as the data on these indicators were collected based on any incidence of violence and harassment in the 12 months preceding the survey. Previous studies have shown that accurately assessing violence against women is difficult due to socio-cultural practices, which may limit or undermine the association between violence and depression [8, 69]. Moreover, our study is cross-sectional, therefore it is not possible to draw a causal inference between depression and the associated factors found in the study. These methodological limitations underscore the need for longitudinal research to establish the causal association between violence and depression among adolescents. On the other hand, the protective effect of years of schooling and other covariates such as mass media and social media exposure did not play a significant role in reducing the depressive symptoms in our study, warranting further investigation in the context of LMICs. Additionally, a thorough investigation into the pathways and potential mediators through which violence and harassment lead to depressive symptoms will be crucial for informing policy decisions and intervention priorities to address the burden of depression. Other factors that might affect depressive symptoms, such as ethnicity, religion, parental education, alcohol intake, BMI, food habits, and family history of depression were not assessed due to the unavailability of data and should be investigated further.

## Policy implications

Our study underscores the need to integrate routine depression screening in primary health care settings that could enable early identification of symptoms among adolescents, as well as timely intervention and support. Moreover, primary health care providers can offer ongoing support and follow-up, ensuring continuity of care for individuals diagnosed with depression to reduce the health and economic burden of this condition. Furthermore, strengthening and revisiting the existing laws to prevent violence and harassment would be crucial for navigating the risk of depression by addressing these modifiable risk factors. In addition, increasing access to health care services, and strict enforcement of laws and policy execution to reduce early marriages are critical for reducing the risk of depression. Moreover, community-based awareness programs can serve as a vital tool in preventing violence and harassment.

## Conclusions

Our analysis demonstrated a high prevalence of depression among married adolescents, although the prevalence of moderately severe and severe depression was relatively low. We identified a substantial number of risk factors associated with depression such as physical harassment, sexual harassment, social bullying, and cyberbullying. To reduce the high prevalence of depression, a community-based routine screening program is crucial to alleviate the burden of depression among married adolescents. Our findings also suggest the need to strengthen the existing laws and their implementation to address the modifiable risk factors such as violence and harassment. Furthermore, community-based interventions that focus on behavioral change through active community engagement can play a critical role in curbing and controlling the ever-evolving forms of violence. Estimating the potential health burden and identifying other modifiable risk factors related to depression could guide policy decisions on the investigation, prevention, and treatment of depressive symptoms among married adolescent girls in Bangladesh.

## Supporting information

**S1 File. Data set—Determinants of depression among ever married adolescent girls in Bangladesh: Evidence from Bangladesh Adolescent Health and Wellbeing Survey 2019–2020.** (XLSX)

**S2 File. Questionnaire.** (PDF)

## Acknowledgments

The authors would like to express their heartfelt gratitude to National Institute of Population Research and Training (NIPORT) and the Health Education and Family Welfare Division of the Ministry of Health and Family Welfare (MOHFW) for making this data publicly available, facilitating its optimal utilization and generating new evidence.

## Author Contributions

**Conceptualization:** Mehedi Hasan.

**Data curation:** Mehedi Hasan, Md Al Amin.

**Formal analysis:** Mehedi Hasan.

**Methodology:** Mehedi Hasan, Md Al Amin.

**Software:** Mehedi Hasan.

**Validation:** Mehedi Hasan, Md Al Amin.

**Writing – original draft:** Mehedi Hasan, Md Al Amin.

**Writing – review & editing:** Mehedi Hasan, Md Al Amin.

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
