## [Decision Letter · Decision Letter 0]

30 Jul 2024

PONE-D-24-04225Determinants of Depression Among Ever Married Adolescent Girls in Bangladesh: Evidence from Bangladesh Adolescent Health and Wellbeing Survey 2019-2020PLOS ONE

Dear Dr. Hasan,

Thank you for submitting your manuscript to PLOS ONE. After careful consideration, we feel that it has merit but does not fully meet PLOS ONE’s publication criteria as it currently stands. Therefore, we invite you to submit a revised version of the manuscript that addresses the points raised during the review process.

**ACADEMIC EDITOR: Please insert comments here and delete this placeholder text when finished.** Be sure to:Indicate which changes you require for acceptance versus which changes you recommendAddress any conflicts between the reviews so that it's clear which advice the authors should followProvide specific feedback from your evaluation of the manuscriptPlease ensure that your decision is justified on PLOS ONE’s publication criteria and not, for example, on novelty or perceived impact.

We look forward to receiving your revised manuscript.

Kind regards,

Md. Shahjalal

Academic Editor

PLOS ONE

Reviewers' comments:

Reviewer's Responses to Questions

**Comments to the Author**

1. Is the manuscript technically sound, and do the data support the conclusions?

Reviewer #1: Partly

Reviewer #2: Partly

2. Has the statistical analysis been performed appropriately and rigorously? 

Reviewer #1: Yes

Reviewer #2: Yes

3. Have the authors made all data underlying the findings in their manuscript fully available?

Reviewer #1: Yes

Reviewer #2: Yes

4. Is the manuscript presented in an intelligible fashion and written in standard English?

Reviewer #1: No

Reviewer #2: No

5. Review Comments to the Author

Reviewer #1: 1. It is a secondary data analysis. Then it should be described so.

2. Introduction can be made more coherent.

3. Table 1 - Check spelling (Syber Bullying)

4. Inconsistent use of abbreviation (IPV not used even after prior elaboration earlier)

5. Statement in 346-347 needs reference

6. Many mistakes in spelling-concrete 6.

7. Meaning of Line 369-379 not clear

8. Line 401 -should be though not through

9. Is it multivariate/univariate analysis? Use standard terms

10. The study is important, but the manuscript needs significant language edits and spell checks

Reviewer #2: Thank you for the opportunity to review the paper titled "Determinants of Depression Among Ever Married Adolescent Girls in Bangladesh: Evidence from Bangladesh Adolescent Health and Wellbeing Survey 2019-2020." The study provides valuable insights into the factors influencing depression in this demographic, utilizing data from a comprehensive national survey. Here is my points.

Introduction

The introduction is informative and thorough. However, streamlining certain sections could enhance readability. You might consider combining similar points and adopting more concise language to improve the flow.

Introducing a theoretical framework could significantly bolster the study's conceptual foundation. For instance, applying the ecological systems theory or the life course perspective might provide valuable insights into the factors influencing depression among married adolescent girls.

To improve clarity and focus, explicitly stating the specific objectives of the study would be beneficial. While the introduction suggests the research aims, clear articulation of these objectives is recommended.

It would be helpful to include precise definitions of key terms such as "depression" and "adolescence." This will ensure consistency and clarity throughout the manuscript.

Enhance the connection between the identified risk factors for depression (e.g., social isolation, limited autonomy) and the specific context of married adolescent girls in Bangladesh.

Address potential ethical considerations pertinent to conducting research with this vulnerable population.

Methods

Considering the complex sampling design, it would be helpful to indicate whether sample weights were applied in the analysis to account for the sampling design and enhance the representativeness of the results.

The exclusion of 383 pregnant women and 394 cases with missing data is mentioned. Providing a more detailed explanation of the missing data mechanisms (e.g., missing completely at random, missing at random, missing not at random) would add clarity.

The information provided on the reliability and validity of the PHQ-9 is not adequate.

Further elaboration on the rationale for selecting the specific covariates would be appreciated. Was the selection guided by a theoretical framework?

Clarifying how missing data for covariates were managed (e.g., imputation, listwise deletion) would strengthen the methodology section.

Discussion

While the discussion provides a good overview of the associations between depression and the studied factors, it could be strengthened by exploring potential mechanisms underlying these relationships. For example, how might physical violence lead to depression?

The discussion could be expanded to provide more specific policy recommendations based on the study findings. What types of interventions could be implemented to address the high prevalence of depression among married adolescent girls?

The authors could elaborate on the potential directions for future research to build upon the findings of this study. For example, longitudinal studies could be conducted to examine the causal relationships between the studied factors and depression.

Consider incorporating a theoretical framework to guide the interpretation of the findings and to explain the potential mechanisms underlying the observed associations.

Discuss the potential role of cultural factors in shaping the mental health experiences of married adolescent girls in Bangladesh.

Provide more specific recommendations for interventions targeting different levels of prevention (primary, secondary, and tertiary) and for different stakeholders (e.g., government, healthcare providers, community organizations).

Conclusion:

While the conclusion calls for policy interventions, it could be strengthened by providing more specific recommendations. For example, the authors could outline potential policy targets, such as strengthening law enforcement against violence, increasing access to mental health services, or promoting gender equality.

The conclusion could end with a strong call to action, emphasizing the urgency of addressing the mental health needs of married adolescent girls.

6. PLOS authors have the option to publish the peer review history of their article (what does this mean?). If published, this will include your full peer review and any attached files.

Reviewer #1: No

Reviewer #2: No

---

## [Author Response · Author response to Decision Letter 0]

20 Sep 2024

Review of manuscript for PLOS ONE: “Determinants of Depression Among Ever Married Adolescent Girls in Bangladesh: Evidence from Bangladesh Adolescent Health and Wellbeing Survey 2019-2020”

Manuscript number: PONE-S-24-05514

# Journal requirements:

Comment: Please ensure that your manuscript meets PLOS ONE's style requirements

Response: Thank you for your concern. We have taken a thorough look on the guideline and checked our formatting and styles in the manuscript as per journal requirements

Comment: Please provide a complete Data Availability Statement in the submission form, ensuring you include all necessary access information or a reason for why you are unable to make your data freely accessible. If your research concerns only data provided within your submission, please write "All data are in the manuscript and/or supporting information files" as your Data Availability Statement.

Response: Thank you for your suggestion. Since the data utilized in this study were extracted from a national survey (secondary source), therefore, we recommend restricting the data access to manuscript submission purposes only, rather than make it freely accessible. We put the recommended statement “All data are in the manuscript and/or supporting information files" in Data Availability Statement.

# Reviewer(s)' Comments to Author:

Reviewer: 1

Overall

Comment: It is a secondary data analysis. Then it should be described so.

Response: We appreciate your valuable feedback. We mentioned it in the introduction section

Comment: Introduction can be made more coherent.

Response: We appreciate your insightful feedback. We try our best to make this section more inclusive and coherent through comprehensive review

Comment: Check spelling (Syber Bullying)

Response: Many thanks, we corrected it accordingly

Comment: Inconsistent use of abbreviation (IPV not used even after prior elaboration earlier)

Response: We appreciate your insightful feedback. We checked this inconsistency and corrected accordingly

Comment: Statement in 346-347 needs reference

Response: Thanks for your insightful feedback. Line 346 indicates the association between sexual violence and depression that found in our study, therefore no reference is needed for this line, on the other hand, for line 347-348, we added the appropriate reference 

Comment: Many mistakes in spelling-concrete 6.

Response: Thanks, we corrected the spelling accordingly

Comment: Meaning of Line 369-379 not clear

Response: Thanks for your excellent observation. Indeed, in our multivariate analysis (AOR- Table-3), education didn’t appear as significant factor in determining the depressive symptoms but appear significant in univariable analysis (COR- Table-3). Therefore, to support and justify our findings, we discussed the relevant literature on the association between education and depression in lines 369-389. In this part of discussion, we have edited few sentences to make it more precise and understandable.

Comment: Line 401 -should be though not through

Response: Thanks, we corrected the spelling accordingly.

Comment: Is it multivariate/univariate analysis? Use standard terms

Response: Thanks for your insightful feedback. In the statistical analyses section, we used the term 'univariable analysis' to represent the Crude Odds Ratio (COR), which does not account for any potential confounders in the model during the analysis (Table 3). On the other hand, we used 'multivariable analysis,' also referred to as 'multivariate analysis,' to indicate the Adjusted Odds Ratio (AOR) in the model, which accounts for potential covariates in the analysis (Table 3). Our findings in the manuscript were interpreted based on this multivariable/multivariate analysis.

Comment: The study is important, but the manuscript needs significant language edits and spell checks

Response: Thanks for your valuable feedback. We put our best efforts to navigate this problem and improve the flow. We have checked thoroughly for settling the grammatical and linguistic errors in our revised paper to make it easier for the readers.

Reviewer: 2

Introduction

Comment: The introduction is informative and thorough. However, streamlining certain sections could enhance readability. You might consider combining similar points and adopting more concise language to improve the flow.

Response: Thanks for your insightful comments. We have given a thorough look to fix all the inconsistencies including linguistics and grammatical mistakes throughout the introduction. Further, we have revised the flow of the introduction to make it easier to understand for the readers.

Comment: Introducing a theoretical framework could significantly bolster the study's conceptual foundation. For instance, applying the ecological systems theory or the life course perspective might provide valuable insights into the factors influencing depression among married adolescent girls.

Response: Thanks for your insightful suggestion. Yes, we follow a theoretical framework in the introduction section to have a solid foundation for conducting this study. The introduction section was systematically guided by a matrix/framework that are as follows: 1. describing the outcome of interest (depression) and its relevance; 2. describing the problem in global and national context as well as highlighted the relevance of the study by stating the available evidence and existing knowledge gap. 3. adopting a theoretical framework based on the socio-ecological model, which indicate individual’s mental health are influenced by multiple levels of indicators among the married adolescents in the context of Bangladesh, including individual, community, and societal (Socio-demographic) factors and discuss accordingly in the introduction section. For instance, we discuss the association between violence and different spectrums of mental health conditions among the married adolescents in Bangladesh. As well as we also discuss the social determinants of depression among the respondents in Bangladesh. 4. Discuss and highlight the significance of the current study by articulating the research gap among the married adolescents 5. Statement of the study objective. 

Comment: To improve clarity and focus, explicitly stating the specific objectives of the study would be beneficial. While the introduction suggests the research aims, clear articulation of these objectives is recommended.

Response: Thanks for your insightful feedback. As recommended, we explicitly stated the specific objectives of the study

Comment: It would be helpful to include precise definitions of key terms such as "depression" and "adolescence." This will ensure consistency and clarity throughout the manuscript.

Response: Thanks for your insightful feedback. Yes, the elaborate and precise definition of these key terms was absent in the manuscript. As suggested, we elaborately defined the “depression, and “adolescence”, according to WHO definition

Comment: Enhance the connection between the identified risk factors for depression (e.g., social isolation, limited autonomy) and the specific context of married adolescent girls in Bangladesh.

Response: Thanks for valuable suggestion. As suggested, we discussed the relevant associations in the introduction section

Comment: Address potential ethical considerations pertinent to conducting research with this vulnerable population.

Response: Thanks for your concern. We conducted this analysis utilizing the national survey data (secondary source) for this study. The original cross-sectional survey obtained ethical approval from the relevant institutional review board and ensured that all respondents or their caregivers provided informed consent before participating in the interviews including the assessment of depression. The details of the ethical considerations, including the participant consent form, were submitted separately during the first submission. However, leveraging this secondary data in our analysis complies with the ethical standards for research involving human subjects, and no additional ethical approval was required for this secondary analysis.

Methods

Comment: Considering the complex sampling design, it would be helpful to indicate whether sample weights were applied in the analysis to account for the sampling design and enhance the representativeness of the results

Response: Thanks for valuable suggestion. Since our study utilized secondary data, we incorporated sampling weights in the regression model (Table-3) to enhance the robustness and generalizability of our findings. As suggested, we have included this information in the 'Statistical Analysis' section of the manuscript

Comment: The exclusion of 383 pregnant women and 394 cases with missing data is mentioned. Providing a more detailed explanation of the missing data mechanisms (e.g., missing completely at random, missing at random, missing not at random) would add clarity.

Response: Thanks for valuable suggestion. In the methodology section, we explicitly addressed the mechanisms of missing data. For instance, a total of 383 observations from pregnant women were excluded purposively due to the known association between depression and pregnancy/early childbearing. Additionally, 394 observations were excluded from the analysis due to missing information on harassment and violence indicators. Given that this missing data was random and unrelated to the outcome of depression, we applied a listwise deletion process to systematically exclude these observations to maintain the integrity of our results.

Comment: The information provided on the reliability and validity of the PHQ-9 is not adequate.

Response: Thanks for valuable suggestion. As suggested, we tried to provide detail information on PHQ-9.

Comment: Further elaboration on the rationale for selecting the specific covariates would be appreciated. Was the selection guided by a theoretical framework

Response: Thanks for valuable recommendations. We were guided by relevant literature from LMICs, that report the determinants of depression among adolescents. Based on this review, we considered the potential covariates for our study. Additionally, we employed a conceptual framework to select these covariates, focusing specifically on socio-demographic factors and indicators of violence, to select the relevant variables.

Comment: Clarifying how missing data for covariates were managed (e.g., imputation, listwise deletion) would strengthen the methodology section.

Response: Thanks for valuable suggestion. Due to 394 missing data points for physical and sexual harassment and other covariates, a listwise deletion process was applied. This approach systematically excluded 394 observations related to depression status to ensure the accuracy and completeness of the analysis. 

Discussion

Comment: While the discussion provides a good overview of the associations between depression and the studied factors, it could be strengthened by exploring potential mechanisms underlying these relationships. For example, how might physical violence lead to depression?

Response: Thanks for your suggestion. Based on available literature, we explicitly discussed the potential pathways for the associations that induce depression among adolescents. Since, there is scarce of information in the context of Bangladesh regarding these pathways, therefore we recommend further research to explore the underlying mechanisms that lead to depression.

Comment: The discussion could be expanded to provide more specific policy recommendations based on the study findings. What types of interventions could be implemented to address the high prevalence of depression among married adolescent girls?

Response: Thanks for insightful recommendation We recommended implementing community-based screening programs for the early detection of depression as part of a primary prevention approach. Additionally, we suggest engaging community healthcare providers to ensure continuity of care reduce this the burden as part of secondary prevention approach. Moreover, we added a section namely “Policy implications” for elucidating these recommendations.

Comment: The authors could elaborate on the potential directions for future research to build upon the findings of this study. For example, longitudinal studies could be conducted to examine the causal relationships between the studied factors and depression.

Response: Thanks. We elaborately discussed this issue in the limitation section. Since we utilized survey/cross-sectional data for this study, that could not establish a causal or dose-response relationship between the risk factors and the outcome of interest (depression). Therefore, we recommend conducting longitudinal or follow-up studies to establish causal associations. Furthermore, we also recommend to explore the potential mediator/pathway of the determinants that later on lead to depression, to fill the knowledge gap and take appropriate interventions.

Comment: Consider incorporating a theoretical framework to guide the interpretation of the findings and to explain the potential mechanisms underlying the observed associations.

Response: We guided by a framework to interpret our findings align with each of the factors that led depression. For example, we explain how various factors—such as socio-demographic characteristics, exposure to violence, and harassment—affect mental health outcomes by considering the context, social norms, and practices in Bangladesh. By integrating this framework, we provided a comprehensive explanation of the pathways that lead to depression, supported by the associations observed in our study. We also discuss the strength and limitations of our observed findings and provide specific recommendations in terms of different prevention approaches.

Comment: Discuss the potential role of cultural factors in shaping the mental health experiences of married adolescent girls in Bangladesh.

Response: We discuss the different dimensions of cultural factors such as such as early marriage, IPV, inequality, poor subjective happiness and limited mobility that shaping and influencing the mental health status in the context of Bangladesh.

Comment: Provide more specific recommendations for interventions targeting different levels of prevention (primary, secondary, and tertiary) and for different stakeholders (e.g., government, healthcare providers, community organizations).

Response: We provided recommendations for preventing and controlling depression, focusing on various prevention mechanisms and engaging with different stakeholders. In addition, we added a section namely “Policy implications” for elucidating these recommendations.

Conclusion

Comment: While the conclusion calls for policy interventions, it could be strengthened by providing more specific recommendations. For example, the authors could outline potential policy targets, such as strengthening law enforcement against violence, increasing access to mental health services, or promoting gender equality.

Response: As recommended, we incorporated this into the conclusion. For example, we recommended introducing the screening program into the community level; enforce the strict law; and behavioral change interventions to prevent any form of violence and harassment

Comment: The conclusion could end with a strong call to action, emphasizing the urgency of addressing the mental health needs of married adolescent girls

Response: This issue is also mentioned in the section as suggested.

---

## [Decision Letter · Decision Letter 1]

8 Nov 2024

Determinants of Depression Among Ever Married Adolescent Girls in Bangladesh: Evidence from The Bangladesh Adolescent Health and Wellbeing Survey 2019-2020

PONE-D-24-04225R1

Dear Mr. Mehedi,

We’re pleased to inform you that your manuscript has been judged scientifically suitable for publication and will be formally accepted for publication once it meets all outstanding technical requirements.

Kind regards,

Md. Shahjalal

Academic Editor

PLOS ONE

Additional Editor Comments (optional):

Reviewers' comments:

Reviewer's Responses to Questions

**Comments to the Author**

1. If the authors have adequately addressed your comments raised in a previous round of review and you feel that this manuscript is now acceptable for publication, you may indicate that here to bypass the “Comments to the Author” section, enter your conflict of interest statement in the “Confidential to Editor” section, and submit your "Accept" recommendation.

Reviewer #3: All comments have been addressed

2. Is the manuscript technically sound, and do the data support the conclusions?

Reviewer #3: Yes

3. Has the statistical analysis been performed appropriately and rigorously? 

Reviewer #3: Yes

4. Have the authors made all data underlying the findings in their manuscript fully available?

Reviewer #3: Yes

5. Is the manuscript presented in an intelligible fashion and written in standard English?

Reviewer #3: Yes

6. Review Comments to the Author

Reviewer #3: (No Response)

7. PLOS authors have the option to publish the peer review history of their article (what does this mean?). If published, this will include your full peer review and any attached files.

Reviewer #3: **Yes: **Dr Md Ahedulla

---

## [Editor Report · Acceptance letter]

13 Nov 2024

PONE-D-24-04225R1 

PLOS ONE

Dear Dr. Hasan, 

I'm pleased to inform you that your manuscript has been deemed suitable for publication in PLOS ONE. Congratulations! Your manuscript is now being handed over to our production team.

Kind regards, 

on behalf of

Dr. Md. Shahjalal 

Academic Editor

PLOS ONE